# Are Mutation Carrier Patients Different from Non-Carrier Patients? Genetic, Pathology, and US Features of Patients with Breast Cancer

**DOI:** 10.3390/cancers14112759

**Published:** 2022-06-02

**Authors:** Roxana Maria Pintican, Angelica Chiorean, Magdalena Duma, Diana Feier, Madalina Szep, Dan Eniu, Iulian Goidescu, Sorin Dudea

**Affiliations:** 1Department of Radiology, Iuliu Hatieganu University of Medicine and Pharmacy, Victor Babes Street, Nr 8, 400347 Cluj-Napoca, Romania; diana.feier@gmail.com (D.F.); madalinaszep@gmail.com (M.S.); sdudea1@gmail.com (S.D.); 2Medimages Breast Center, Brancusi Street, Nr 133, 400462 Cluj-Napoca, Romania; magdaduma@gmail.com; 3Ion Chiricuta Oncology Institute, Republicii Street, Nr 34–36, 400015 Cluj-Napoca, Romania; daneniu@yahoo.com (D.E.); iuliangoidescu@gmail.com (I.G.); 4Department of Surgery, Iuliu Hatieganu University of Medicine and Pharmacy, Louis Pasteur Street, Nr 4, 400012 Cluj-Napoca, Romania

**Keywords:** high-risk population, *BRCA*, benign appearance breast cancer, *CHEK*, US

## Abstract

**Simple Summary:**

Up to 30% of breast cancer patients are carriers of pathogenic mutations in breast cancer susceptibility genes. Except for *BRCA1/2* genes that account only for 22–30% of hereditary breast cancer, less is known about the remaining genes that are prone to breast cancer. Our aim was to retrospectively evaluate the relationship between the pathogenic mutations (*BRCA* and non-*BRCA*), US features, and histopathologic findings of breast cancer patients with and without mutations. We concluded that carrier patients (*BRCA*, *TP53*, *PALB*, *CHEK*, *ATM*, *RAD*) seem to exhibit benign imaging findings on US compared to mutation-negative patients. Furthermore, carrier patients had the majority of tumors with higher histologic grade and a higher proliferation index. *BRCA1*, *TP53*, and *RAD* carriers accounted for up to one third of the ER-negative tumors from the mutation group. Axillary US performed worse in depicting axillary metastatic lymph nodes in carrier patients, compared to negative patients.

**Abstract:**

The purpose of this study is to evaluate the relationship between the pathogenic/likely pathogenic mutations, US features, and histopathologic findings of breast cancer in mutation carriers compared to non-carrier patients. Methods: In this retrospective study, we identified 264 patients with breast cancer and multigene panel testing admitted to our clinic from January 2018 to December 2020. Patient data US findings, US assessment of the axilla, multigene panel tests, histopathology, and immunochemistry reports were reviewed according to the BI-RADS lexicon. Results: The study population was comprised of 40% pathogenic mutation carriers (*BRCA1*, *BRCA2*, *CHEK2*, *ATM*, *PALB*, *TP 53*, *NBN*, *MSH*, *BRIP 1* genes) and 60% mutation-negative patients. The mean patient age was 43.5 years in the carrier group and 44 years in the negative group. Carrier patients developed breast cancer with benign morphology (acoustic enhancement, soft elastography appearance) compared to non-carriers (*p* < 0.05). A tendency towards specific US features was observed for each mutation. *BRCA1* carriers were associated with BC with microlobulated margins, hyperechoic rim, and soft elastography appearance (*p* < 0.05). Estrogen receptor (ER)-negative tumors were associated with *BRCA1*, *TP53*, and *RAD* mutations, while *BRCA2* and *CHEK2* were associated with ER-positive tumors. Conclusions: Patients with pathogenic mutations may exhibit BC with benign US features compared to negative, non-carrier patients. *BRCA1*, *TP53*, and *RAD* carriers account for up to one third of the ER tumors from the carrier group. Axillary US performed worse in depicting involved lymph nodes in carrier patients, compared to negative patients.

## 1. Introduction

There are 12 established breast cancer-predisposition genes that are known to have an increased risk of developing cancer [1]. Up to 30% of breast cancer patients are positive for cancer-predisposition genes, evaluated through hereditary multigene testing panels. 

Compared to *BRCA1/2* mutation carriers, which account for up to 22–30% of the hereditary breast cancer cases, less is known about the other 70% of genetic breast cancer patients [2]. There is an increasing number of studies reporting differences in terms of natural history and treatment response related to each pathogenic mutation. For example, the *ATM* carrier patients are more prone to develop subcutaneous necrosis and contralateral breast cancer after radiotherapy, which may be a relative contraindication to the standard management [3,4]. These aspects support the idea of different breast cancer sub-types being linked to each mutation and may also be reflected in different breast cancer imaging features. However, except for the *BRCA1/2* genes, there are limited studies regarding the other breast-cancer susceptibility genes, on what breast cancer type these patients are prone to, their imaging features, or histopathology characteristics. 

Currently, the NCCN recommend mastectomy in breast cancer patients with *BRCA* mutations and suggest that mastectomy could be offered in high/moderate-risk patients with other pathogenic/likely pathogenic mutations [5]. *BRCA1/2*, *PALB2*, *TP53*, *PTEN*, *CDH1*, and *STK11* are high penetrant genes, associated with an increased risk of developing breast cancer of >60%, while *ATM*, *CHEK2*, and *RAD* are moderate penetrant genes with a breast cancer risk of 40–60%. Moreover, the American Society of Surgeons recommends genetic testing in all breast cancer patients [6]. Thus, the number of mutation carriers will continue to rise, increasing the need for informed, familiarized breast imagers with their tumor features and possible misdiagnosis characteristics.

Therefore, our study aims to evaluate and compare the relationship between US features and pathologic findings (including aggressiveness markers, molecular type, and axillary lymph node status) of breast cancer that develop in patients who are carriers of pathogenic mutations, versus negative, non-carrier patients. 

## 2. Material and Methods

### 2.1. Study Population

This retrospective study was approved by the institutional review board of BLINDED* (SR NR 9, from 18 January 2021) and the need for informed consent was waived. Inclusion criteria represented patients with breast cancer, pre-operative breast and axillary US, hereditary multigene testing panels, complete surgery, and pathology reports who presented to our clinic between January 2018 and December 2020. 

Exclusion criteria consisted of patients with inadequate or incomplete US images, pathology reports, or genetic tests. Generally, we recommend genetic testing if one of the following criteria are met: breast cancer diagnosed <35 years, bilateral cancer, triple-negative sub-type, one first-degree relative diagnosed with breast cancer <55 years, two second-degree relatives diagnosed with breast cancer <55 years, and if additional melanoma, colon, pancreas, or ovarian cancer is present. Multigene panel testing is a type of genetic testing which analyzes mutations in multiple genes at once. Furthermore, it provides a better understanding of cancer risk compared to single-gene testing. 

Up to 48% of the carrier patients and 43% of the mutation-negative patients exhibited symptoms. The remaining presented for breast US by means of an opportunistic screening since there is no organized governmental breast cancer screening in Romania.

A total of 309 patients were identified, and 98 pathogenic mutation carriers and 145 mutation-negative patients were included in the study (Figure 1).

Patient data including symptoms at the time of diagnosis were also recorded. 

### 2.2. Imaging Technique

Breast US was performed by one radiologist with more than 15 years of experience in breast imaging, using a Hi Vision Ascendus (Hitachi Ltd., Tokyo, Japan) machine with a Wide-Band (6.5–13 MHz) linear probe, and Hologic Supersonic (Aixplorer Mach 30, Aix-en-Provence, France) with a linear probe (5–18 MHz). Greyscale, Color Doppler, and strain elastography (SE) images were retrieved and interpreted by two radiologists with 15 and 4 years of experience and consensus was reached for discordant cases. The two radiologists were blinded to any existing mammography or MRI exams. Greyscale and Color Doppler features were described using the American College of Radiology BI-RADS lexicon (5th edition) [7]. Circumscribed margins, parallel orientation compared to the skin, and posterior enhancement were considered “benign US features”. Microlobulated, spiculated, or indistinct margins, taller than wide orientation, and posterior acoustic shadowing were categorized as “suspicious US features”. The mass homogeneity (homogeneous, heterogeneous) and echogenicity (hypo-, iso-, hyper-echoic, or mixed) were also assessed. The presence of microcalcifications suspected on US was confirmed with mammography for all cases. SE images were classified according to the Ueno–Itoh adaptation of the Tsukuba elasticity score, considering ACR Appropriateness Criteria [8]. All patients with suspect axillary lymph nodes at US (absent fatty hilum, cortical thickness >3 mm, indistinct contour) were considered abnormal on imaging and underwent US-guided core needle biopsy.

### 2.3. Pathologic and Genetic Data

Pathologic data were reviewed, including the histologic tumor type, size, histologic grade, lymph node status, and immunohistochemistry findings (estrogen and progesterone receptors—ER, PR, HER2 status, ki-67% proliferation index). 

Multigene panel testing, including 12 established breast cancer-predisposition genes (*BRCA1*, *BRCA2*, *TP53*, *ATM*, *CHEK2*, *PALB 2*, *BARD 1*, *NBN*, *MSH*, *RAD 51C* and *RAD 51D*, *BRIP 1*), were classified as pathogenic or likely pathogenic according to the ClinVar database and included in the carrier group. Patients positive only for variants of uncertain significance (VUS) were included together with patients that tested negative for all panel genes, in the mutation-negative group.

### 2.4. Statistical Analysis

Statistical analyses were performed using MedCalc software (version 19.2.6, Ostend, Belgium). To analyze the associations among mutations, clinic-pathologic data, and US findings, the Chi-square or Fisher’s exact test was used. Moreover, the Mann–Whitney U test was used to compare the age, lesion size, and elastography score between patients with and without pathogenic mutations. The agreement between US and surgery in the detection of axillary lymphadenopathy was calculated for each group. *p* < 0.05 was considered to indicate a statistically significant difference between groups.

## 3. Results

The carrier group consisted of 98 pathogenic mutation patients divided as follows: 29 *BRCA1*, 15 *BRCA2*, and 62 *non-BRCA1/2* (15 *CHEK2*, 15 *RAD 51C* or *D*, 7 *PALB 2*, 6 *NBN*, 3 *TP 53*, 3 *ATM*, 2 *BARD 1*, 2 *MSH 2*, 1 *BPRIP 1*). The mean patient age was 43.5 years (range 30–67 years) in the carrier group and 44 years (range 24–73 years) in the mutation-negative group.

### 3.1. Associations between Clinico-Pathological Data and Mutation Status

At the time of diagnosis, 47 of 98 (48%) of the carrier group patients and 62 of 145 (43%) mutation-negative patients exhibited symptoms. The remaining presented for breast US by means of an opportunistic screening since there is no organized governmental breast cancer screening in Romania. For both groups, the palpable breast mass was the most common symptom (35/98, 46/145), and only few patients had nipple discharge (10/98, 16/145). In particular, two patients with *BRCA1* and *CHEK2* mutations presented only with axillary discomfort (2/98, 2%). No significant difference was observed for patient provenience, or symptoms at the time of diagnosis (all *p* > 0.05).

Invasive ductal carcinoma of no special type (IDC-NST) was the most common type of breast cancer in both groups. The tumor size, lymphadenopathy, lympho-vascular invasion, ER, and HER2 status did not significantly differ (all *p* > 0.05).

The carrier group had a significantly higher number of unifocal tumors, with higher histologic grade, and higher proliferative index ki-67% (*p* = 0.03, *p* < 0.000, and *p* < 0.001). Additional VUS were found to be associated with 24 out of 98 mutation carrier patients (*p* < 0.0001) (Table 1).

### 3.2. Associations between US Features and Mutation Status

Upon US, the dominant finding was breast mass in both carrier (93%) and mutation-negative (97%) groups, and only few patients exhibited a non-mass appearance (7% and 3%, respectively). No significant difference was observed for shape or orientation (all *p* > 0.05), with irregular and non-parallel masses as the most frequent finding in both groups (Figure 2). 

Margins, echo patterns, and posterior features were found to be statistically different between carrier and mutation-negative groups (*p* = 0.047, *p* < 0.0001, and *p* < 0.0001). Microlobulated margins were found in 10% of the carrier patients and only in 5% of the mutation-negative patients (*p* = 0.23) (Figure 3). 

The heterogeneous echo pattern and acoustic enhancement were associated with pathogenic mutation carriers (*p* < 0.0001, <0.0001), while spiculated margins, hypoechoic pattern, and posterior acoustic shadowing were associated with mutation-negative patients (*p* = 0.047, <0.0001, and <0.0001) (Figure 4).

The presence of calcifications was associated with the carrier group (*p* = 0.001) and there was a significant difference between the two groups with regard to the calcification type (Figure 5). Consequently, calcifications associated with a mass (30/35) and calcifications alone (3/35) were associated with carriers, while calcifications within ducts (6/23) were associated with mutation-negative patients (*p* = 0.001). 

Hyperechoic rim was found to be associated with the carrier group (*p* = 0.001), while the mutation-negative group had no associated features in the majority of cases. 

No statistical difference was found regarding the Color Doppler signal and the BI-RADS category between the two groups (all *p* > 0.05).

SE was statistically different between groups and showed a soft appearance with lower scores (2, 3, or blue-green-red appearance) in patients with mutations and a hard appearance with higher scores (4, 5) in mutation-negative patients (*p* = 0.029) (Figure 6).

Axillary US was positive in 37 out of 98 pathogenic mutation carriers and 77 out of 145 negative patients. For the carrier group, the positive axillary US cases corresponding to positive axillary surgery reached a moderate agreement (kappa = 0.48, *p* = 0.000). For the mutation-negative group, US corresponding to surgery reached a substantial agreement (kappa = 0.656, *p* < 0.0001) in depicting axillary lymph node involvement (Table 2). 

### 3.3. Associations between US Features and Specific Pathogenic Mutations

ER-negative tumors were associated with *BRCA1* patients (*p* < 0.0001) and *TP53* patients (*p* = 0.002) and were also found in the *RAD* mutations group (33% of tumors, *p* = 0.36). ER-positive tumors were associated with *BRCA2* (*p* = 0.038) and *CHEK2* (*p* = 0.038) carriers and were also found in *PALB2* (85% of tumors, *p* = 0.15), *NBN* (100% of tumors, *p* = 0.18), and *ATM* (100% of tumors, *p* = 0.3) patients.

Breast masses with circumscribed margins, hyperechoic rim, and soft elastography appearance were associated with *BRCA1* mutations (*p* < 0.0001, 0.000, and 0.05, respectively) (Figure 6). No statistical difference was noted between patients with other pathogenic mutations, compared to the negative group (all *p* > 0.05). The pathologic characteristics and US features seen in more than half of each genetic mutation sub-group are summarized in Table 3.

## 4. Discussion

In the current study, we found that breast cancer’s histology and US features of pathogenic mutation carriers differ from the mutation-negative patients. Mutation carriers (*BRCA* and non-*BRCA*) tend to develop breast cancer with benign morphologic features and more aggressive pathologic characteristics. *BRCA1*, *TP53*, and *RAD* pathogenic mutation carriers account for a large percentage (28.5%) of ER-negative tumors. 

Up to 30% of the invasive cancers seen in the high-risk population exhibited benign findings, with round- or oval-shaped masses and smooth margins [8,9,10,11]. Kuhl et al. [12] reported that up to 38% of the genetic breast cancer exhibited benign mammography, US, and MRI features with hypo/anechoic masses with circumscribed margins and parallel orientation. However, out of 13 cancers, 7 had benign features and only 5 of them were *BRCA*-positive. In contrast, a study reported exclusive malignant phenotypes for 20 *BRCA*-associated breast cancers, in all imaging modalities [13]. In our study, we had 44 *BRCA*-associated breast cancers and 54 non-BRCA pathogenic mutation-associated cancers, which exhibited benign morphologic features compared to the 145 mutation-negative breast cancer patients. This could lead to misdiagnosis or delayed diagnosis in these particular patients. Thus, heterogeneous echo pattern, acoustic enhancement, and hyperechoic rim were associated with the pathogenic group. Moreover, we found that spiculated margins, hypoechoic pattern, and posterior acoustic shadowing were features associated with mutation-negative patients (all *p* < 0.05). The morphologic features might be linked to the presence of pathogenic mutations which led to the development of highly aggressive tumors, compared to negative patients. In relation, we found that carrier patients were associated with a higher histologic grade and a higher proliferative index (all *p* < 0.05) compared to the mutation-negative group. 

High-grade cancers exhibit benign features due to their rapid growth, whereas low/intermediate cancers develop a desmoplastic reaction and appear as spiculated masses [14]. Heterogeneous echo pattern and acoustic enhancement might be secondary to the cystic necrotic areas found within rapid growth tumors and may occasionally be misinterpreted as benign lesions. Hyperechoic rim is usually associated with benign pathology, such as pseudo-angiomatous stromal tumors and myoblastic tumors, being caused by inflammatory peritumoral cells [15]. This particular US aspect may contribute and partially correspond to the MRI rim enhancement reported in these high-risk patients [16].

The presence of calcifications with an accompanying mass was associated with the carrier group (*p* = 0.001) and found in 30.6% of the lesions. Intraductal calcifications were associated with the mutation-negative group (*p* = 0.001), corresponding to their higher number of DCIS cases. The high incidence of associated calcifications was previously reported for *BRCA1* and *BRCA2* patients [11,17], and furthermore suggests that mammography remains an important screening and diagnostic imaging modality in all genetic carrier patients.

The high-grade tumors’ appearance on elastography is still a matter of debate. Ye et al. reported significantly softer high-grade tumors compared to low-grade ones (*p* < 0.001), while Ganau et al. reported no statistical difference in high- versus low-grade tumors [17,18]. We found that soft SE appearance was associated with the carrier group (*p* = 0.029). Additionally, the BGR appearance that was previously reported in benign, cystic lesions, was noted in few ER-negative tumors (9, 9% in carriers, versus 1, 0.001% in non-carriers). The first explanation may be found in the presence of abundant necrosis, with a predominant cystic component, seen in these highly aggressive tumors. In addition, in our study, 10% of the tumors were of special sub-types, such as mucinous-, medullary-, or adenoid cystic-type tumors, which may have a minor solid component that could further contribute to this appearance [19,20]. Therefore, we suggest that the BGR aspect seen in solid tumors be re-named as “false BGR” appearance.

As regards to axillary lymphadenopathy, we observed a moderate agreement between axillary US and surgery in carrier patients (kappa = 0.48, *p* = 0.000) and substantial agreement in mutation-negative patients (kappa = 0.656, *p* = 0.000). The high histologic grade leading to an increased number of lymph node micro-metastases (<2 mm) could explain the discrepancy in agreement. Our findings advocate that genetic carrier patients are more prone to have false-negative axillary US compared to mutation-negative patients. However, recent data suggest different immunohistochemistry factors to be involved in axillary metastasis in carrier and mutation-negative patients [21]. 

Up to 24.4% of the pathogenic mutation carriers were also positive for VUS genetic changes. A recent population-based study reported up to 19% of VUS among patients and controls [1]. It remains questionable whether additional VUS have an impact on treatment response or patients’ prognosis, having an uncertain role in the breast cancer pathogenesis.

We performed a separate analysis on each pathogenic mutation group of patients and observed a tendency towards specific US features, found in more than half of the tumors from each subgroup. However, except for the *BRCA1* group, no statistical difference was noted among patients, mainly because of the low numbers of cancers (all *p* > 0.05). To the best of our knowledge, no imaging features were previously reported for non-*BRCA* mutation-associated breast cancers.

Imaging findings of *BRCA*-associated breast cancer were previously reported and seem to exhibit benign features on MRI and mammography [8,9,10,11]. One particular study showed the US features of BRCA-associated tumors, which were primarily hypoechoic masses with irregular shape, parallel orientation, and non-circumscribed margins [11]. The same authors compared *BRCA1* with *BRCA2* tumors and reported a tendency towards benign morphology in *BRCA1* patients, due to their association with acoustic enhancement. We found additional benign morphologic features, such as circumscribed margins, hyperechoic rim, and soft elastography appearance, that were associated with *BRCA1* patients (all *p* < 0.005) compared to mutation-negative patients. 

Additionally, we found that *BRCA1*, *RAD51C*, *RAD51D*, and TP53 carriers were associated with ER-negative tumors, while *BRCA2*, *CHEK*, and *ATM* carriers developed exclusively ER-positive tumors. Our findings are in agreement with one recently published study [1]. 

This study has some limitations. First, the study was retrospective and included consecutive genetically tested patients; however, radiologists were aware that a tumor was present and that would have had possible implications on the BI-RADS category assessment. Second, we included only US as an imaging modality. We are a tertiary referral center, where the patients present mainly for biopsy and pre-surgical planning. Thus, mammography and MRI are usually available and evaluated for a second opinion. There were multiple difficulties that prevented us from including them in the current study (e.g., different mammography and MRI machines, different MRI protocols, legal-related policies). Third, for some pathogenic mutations, the number of patients was limited, related to their low incidence. Finally, our study was based on a single institution. A larger, population-based study will be needed in the future to validate our findings. 

## 5. Conclusions

Patients with pathogenic mutations in *BRCA* and non-*BRCA* genes (such as *TP53*, *PALB*, *CHEK*, *ATM*, and *RAD*) seem to exhibit benign imaging findings on US compared to mutation-negative patients. It remains questionable if the aggressiveness markers, such as high histologic grade, high proliferation index, or ER-negative status, rather than the presence of mutation, tend to exhibit benign morphologic features. *BRCA1*, *TP53*, and *RAD* carriers accounted for up to one-third of the ER-negative tumors from the mutation group. Axillary US performed worse in depicting axillary metastatic lymph nodes in these patients, compared to the mutation-negative patients. 

## Figures and Tables

**Figure 1 cancers-14-02759-f001:**
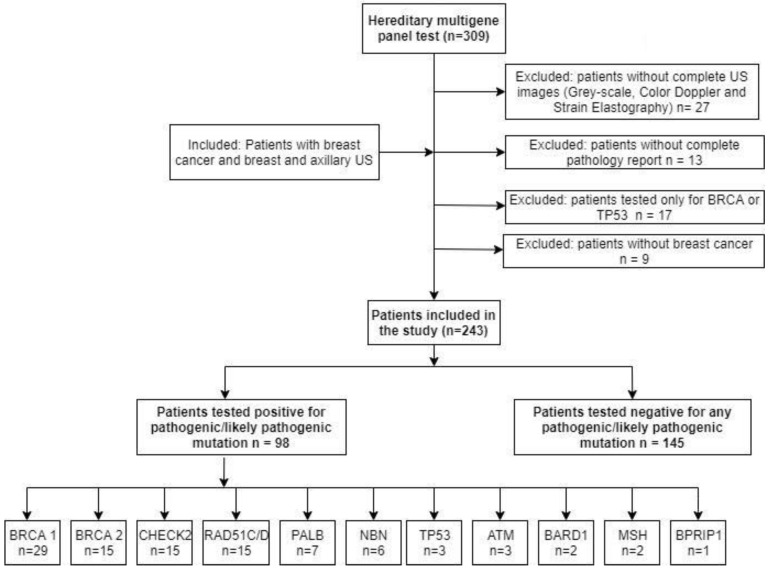
Study population—inclusion and exclusion criteria.

**Figure 2 cancers-14-02759-f002:**
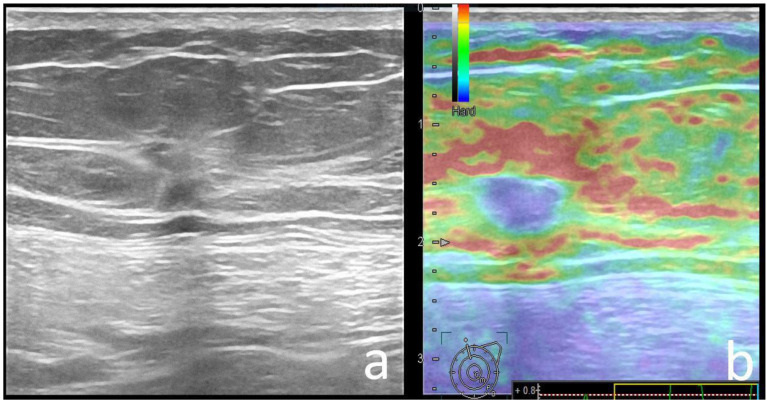
60-year-old *BRCA2* mutation carrier patient with left breast cancer. There is an irregular, slightly hypoechoic mass with indistinct margins and non-parallel orientation compared to the skin (**a**), with a hard elastography appearance (TSUKUBA score 5, **b**). Pathology: IDC-NST, ER/PR-positive, HER2-negative, grade 2, ki67 = 25%.

**Figure 3 cancers-14-02759-f003:**
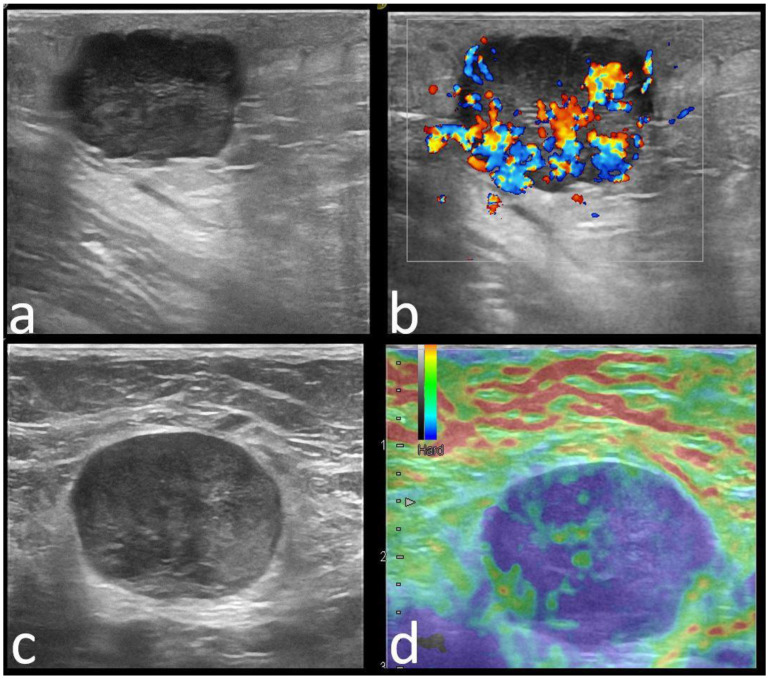
Upper images: 29-year-old *TP53* mutation carrier patient with right breast cancer. There is an oval, hypoechoic mass with microlobulated margins, parallel orientation compared to skin, with acoustic enhancement (**a**) and internal vascularity (**b**). Skin invasion was suspected and confirmed later by pathology. Pathology: IDC-NST, ER/PR/HER2-negative, ki67 = 90%. Lower images: 62-year-old *PALB2* mutation carrier patient with right breast cancer. There is an oval, hypoechoic mass with circumscribed margins, parallel orientation compared to skin (**c**), mild acoustic enhancement, and soft elastography appearance (TSUKUBA score 3, **d**). Pathology: IDC-NST, ER/PR-negative, HER2-positive, grade 2, Ki67 = 80%.

**Figure 4 cancers-14-02759-f004:**
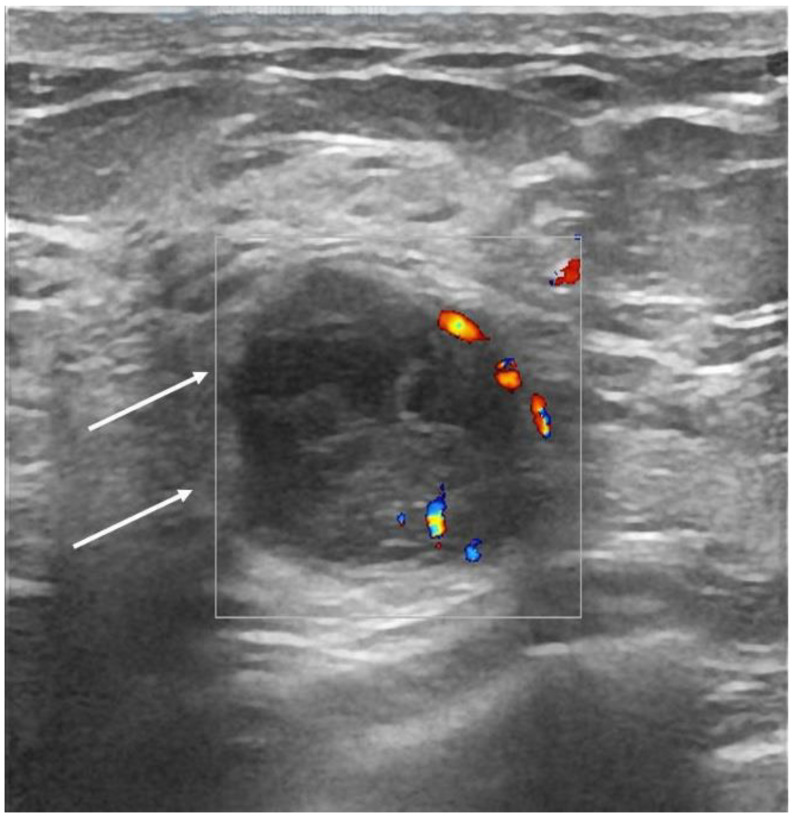
42-year-old *RAD51C* mutation carrier patient with right breast cancer. There is an oval, hypoechoic mass with indistinct margins (arrows), parallel orientation compared to skin, with internal vascularity and soft elastography appearance (TSUKUBA score 2). Pathology: IDC-NST, ER/PR/HER2-negative, grade 3, Ki67 = 80%.

**Figure 5 cancers-14-02759-f005:**
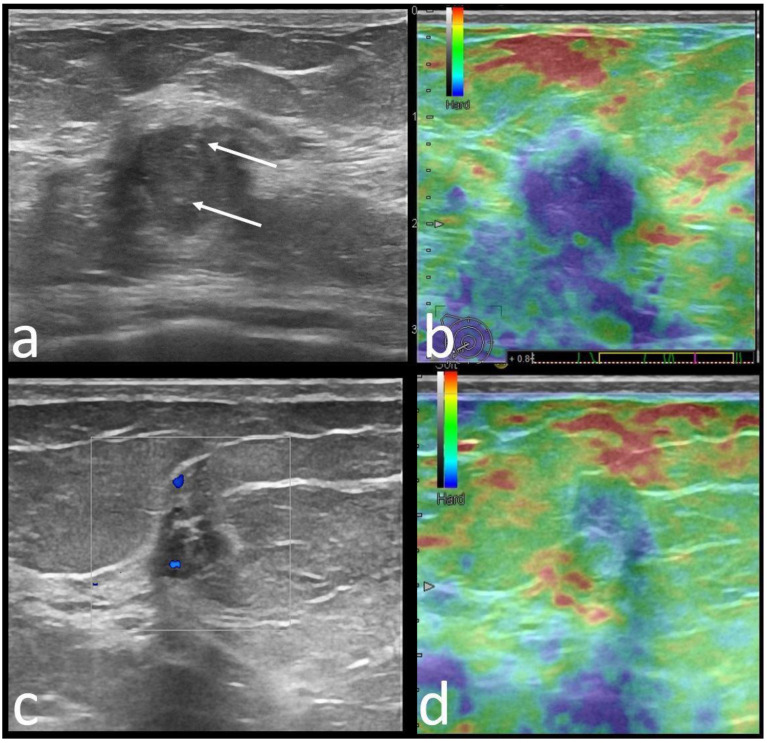
Upper images: 66-year-old ATM mutation carrier patient with right breast cancer. There is an irregular, isoechoic mass with indistinct margins, non-parallel orientation compared to skin (**a**), and soft elastography appearance (TSUKUBA score 3, **b**). Two punctate microcalcifications are seen within the mass (arrows), confirmed on mammography (not shown). Pathology: IDC-NST, ER/PR-positive, HER2-negative, Ki67 = 14%. Lower images: 43-year-old CHEK2 mutation carrier patient with right breast cancer. There is an irregular, heterogeneous mass with indistinct margins, parallel orientation compared to chest wall, with mild posterior acoustic shadowing, internal vascularity (**c**), and soft elastography appearance (**d**, TSUKUBA score 2). Pathology: IDC-NST, ER/PR-positive, HER2-negative, Ki67 = 13%.

**Figure 6 cancers-14-02759-f006:**
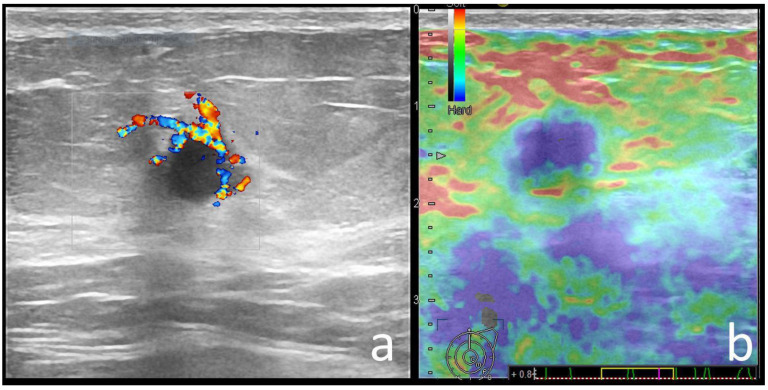
53-year-old *BRCA1* mutation carrier patient with left breast cancer. There is a round, circumscribed, hypoechoic mass, with periphery vessels (**a**) and a blue-green-red elastography appearance (**b**). Pathology: IDC-NST, ER/PR/HER2-negative, grade 3, ki67 = 90%.

**Table 1 cancers-14-02759-t001:** Clinic-pathological characteristics of the patients.

Variable	Pathogenic Carrier Group	Mutation-Negative Group	*p*-Value
Patient age (y), mean (range)	43.5 (30–67)	44 (24–73)	0.644
Patient origin			0.957
Urban	80 (81.6)	120 (82.8)	
Rural	18 (18.4)	25 (17.2)	
Symptoms			0.424
Absent	51 (52)	83 (57)	
Present	47 (48)	62 (43)	
Breast cancer type			
Invasive ductal carcinoma NST	88 (89.8)	120 (82.8)	0.178
Other *	10 (10.2)	25 (17.2)	
“In Situ” component	13 (13.3)	29 (20)	0.234
Number of tumors			**0.032**
Unifocal	61 (62.2)	67 (46.2)	
Multifocal	14 (14.3)	37 (25.5)	
Multicentric	23 (23.5)	41 (28.3)	
Tumor size (mm), mean (range)			0.884
<2 cm	24 (24.5)	29 (20)	
>2 cm	74 (75.5)	116 (80)	
Lymph node status			0.329
Negative	45 (46.9)	78 (54.2)	
Positive	51 (53.1)	66 (45.8)	
Histologic grade			**0.000**
Low	6 (6.1)	33 (22.8)	
Intermediate	49 (50)	80 (55.2)	
High	43 (43.9)	32 (22.1)	
Lympho-vascular invasion			0.927
Absent	72 (73.5)	105 (72.4)	
Present	26 (26.5)	40 (27.6)	
Immunohistochemistry			
ER+	68 (69.4)	110 (75.8)	**0.3**
ER−	30 (30.6)	35 (24.1)	
HER2+	21 (21.4)	23 (15.8)	
HER2−	77 (78.5)	122 (84.1)	**0.3**
Ki-67% status			**0.001**
>20%	77 (78.5)	84 (60)	
<20%	21 (21.4)	61 (42)	
TNM Stage			**0.002**
0	4 (4.1)	3 (2.1)	
I	10 (10.2)	21 (14.5)	
IIA	38 (38.7)	59 (40.7)	
IIB	22 (22.4)	30 (20.7)	
IIIA	15 (15.3)	27 (18.6)	
IIIC	8 (8.2)	3 (2.1)	
IV	1 (1)	2 (1.4)	
Variants of uncertain significance (VUS)			**0.000**
Yes	24 ^a^ (24.4)	4 (2.7)	
No	74 (75.5)	141 (97.2)	

* Includes mucinous, metaplastic, papillary carcinoma, and adenoid cystic. ^a^ Except for one BRCA1 patient who had concomitant pathogenic CHEK2 and PALB2 mutations and CDH1 VUS, the rest of the patients had two concomitant genetic changes.

**Table 2 cancers-14-02759-t002:** Breast cancer US features in carrier and non-carrier patients.

US Feature	Pathogenic Carrier Group	Negative, Non-Carrier Group	*p*-Value
Lesion type			0.107
Mass	91 (93)	141 (97)	
Non-mass	7 (7)	4 (3)	
Shape			0.391
Round	6 (6.1)	14 (9.7)	
Oval	27 (27.6)	31 (21.4)	
Irregular	65 (66.3)	100 (69)	
Orientation			0.861
Parallel	44 (44.9)	68 (46.9)	
Non-parallel	54 (55.1)	77 (53.1)	
Margins			
Circumscribed	22 (22.4)	44 (30.3)	0.226
Non-circumscribed			
Spiculated	19 (19.4)	46 (31.7)	**0.047**
Indistinct			
Angular	21 (21.4)	19 (13.1)	**0.263**
Microlobulated	10 (10.2)	8 (5.5)	
Echo pattern			**0.000**
Hypoechoic	57 (58.2)	100 (70)	
Heterogeneous	36 (36.7)	19 (13)	
Isoechoic	5 (5.1)	26 (17)	
Posterior features			**0.000**
None	39 (39.8)	71 (49)	
Enhancement	27 (27.6)	12 (8.3)	
Shadowing	15 (15.3)	54 (37.2)	
Combined	17 (17.3)	8 (5.5)	
Calcifications			**0.001**
Absent	63 (64.3)	122 (84.1)	
Present *	35 (35.7)	23 (15.9)	
A	30 (30.6)	15 (10.3)	
B	3 (3.1)	2 (2.1)	
D	2 (2)	5 (3.4)	
Associated features			**0.001**
None	59 (60.2)	115 (79.3)	
Hyperechoic Rim	11 (11.2)	8 (5.5)	
Duct ectasia	10 (10.2)	16 (11)	
Distortion	6 (4.1)	18 (18.4)	
Color Doppler signal			0.696
Absent	9 (9.3)	17 (11.7)	
Present	88 (90.7)	128 (88.3)	
Strain Elastography			**0.029**
Soft	40 (40), 9 BGR	30 (20.6), 1 BGR	
Hard	58 (60)	115 (79.3)	
BI-RADS			0.799
4	12 (12.2)	15 (10)	
5	86 (87)	130 (90)	
Axillary US			**0.026**
Negative	61 (62.2)	68 (46.9)	
Positive	37 (37.8)	77 (53.1)	

* A = Calcifications within mass, B = calcifications without mass, D = intraductal calcifications. Soft elastography score = 1, 2, 3 or blue-green-red appearance. Hard elastography appearance = 4 or 5 score. BGR = blue-green-red strain elastography appearance.

**Table 3 cancers-14-02759-t003:** Pathologic characteristics and US features of breast cancer patients associated with pathogenic mutations.

Variable	BRCA1	BRCA2	CHEK2	RAD Group *	PALB	NBN	TP 53	ATM
No. of patients (%)	29 (29.5)	15 (15.3)	15 (15.3)	15 (15.3)	7 (7.1)	6 (6.1)	3 (3)	3 (3)
Breast cancer type	IDC-NST **	IDC-NST	IDC-NST	IDC-NST	IDC-NST	IDC-NST	IDC-NST	IDC-NST
Molecular sub-type (%)	**ER−** (69)	ER+ (100)	ER+ (100)	ER+ (66) **ER−** (33)	ER+ (85)	ER+ (100)	**ER−** (100)	ER+ (100)
US features (+/No. of patients) Orientation	NP (17/29)	NP (9/15)	NP (10/15)	P (7/15)	P (4/7)	P (4/6)	P (2/3)	P (2/3)
Margins	C (20/29)	NC (11/15)	NC (11/15)	NC (11/15)	C (5/7)	NC (5/6)	C (3/3)	NC (3/3)
Echo pattern	Hypoechoic (19/29)	Hypoechoic (9/15)	Heterogeneous (5/15)	Hypoechoic (11/15)	Hypoechoic (4/7)	Heterogenous (3/3)	-	Hypoechoic (2/3)
Posterior features	Enhancement (16/29)	Absent (9/15)	Shadowing (6/15)	Enhancement (6/15)/Combined pattern (5/15)	No posterior (4/7)	-	Enhancement (2/3)	-
Associated features	Hyperechoic rim (6/29) Soft elastography (11/29)	Calcifications (7/15) Hard elastography (12/15)	Calcifications (7/15)	-	-	Calcifications (3/6) Hyperechoic rim (3/6)	-	Calcifications (2/3) Architectural distortion (2/3)

* RAD51C, RAD51D; ** IDC-NST = invasive ductal carcinoma no special type; ER+ = estrogen receptor-positive; ER− = estrogen receptor-negative; P = parallel, NP = non-parallel; C = circumscribed margins, NC = non-circumscribed margins. “-” = No predominant US feature.

## Data Availability

The data presented in this study is available on request from the corresponding author.

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
