# Peer review of "Are Mutation Carrier Patients Different from Non-Carrier Patients? Genetic, Pathology, and US Features of Patients with Breast Cancer"

_cancers, 2022, doi:10.3390/cancers14112759_

Round 1
Reviewer 1 Report
Authors present a work addressing: ‘Are mutation carrier patients different from non-carriers patients? - genetic, pathology and US features of patients with breast cancer’. The aim of the study was to evaluate the relationship between the pathogenic/likely pathogenic mutations, ultrasound features and histopathologic findings of breast cancer (BC) in mutation carriers compared to non-carrier patients. The general conclusions demonstrate that patients with pathogenic mutation may exhibit BC with benign ultrasound features compared to negative, non-carriers patients. BRCA1, TP53 and RAD carriers account for up to one third of the ER-tumors from the carriers group. Axillary ultrasound performed lower in depicting involved lymph nodes in carrier patients, compared to negative patients. The topic of the article is relevant for clinical practice. However, the paper presents a few major issues including:
Generał
1. Interestingly, there is lack of authors name and affiliation.
2. Minor modification of the grammar and punctuation is required.
3. I suggest to avoid abbreviation in the topic of publication.
4. The manuscript is not prepared according journal guidelines: https://www.mdpi.com/journal/ijms/instructions.
Major
1.Please correct p-Values from p=.000 into P<0.0001.
2. Please separate inclusion and exclusion criteria into separate paragraph.
3. In the first paragraph of the results authors should provide short communicate related to detailed patients characteristic.
4. Multigene panel testing should be presented form methodological perspective.
General: interesting, well-conducted work.
Reviewer 2 Report
The abbrevation in line 17 "ER-" is not specified. Moreover, the abbrevation "BC" is used only in the abstract and not thereafter.
Genes are often written incorrectly and not in italics (for example line 11 "BRCA 2" instead of "BRCA2", line 17 "RAD" instead of "RAD50" or "RAD51C" or "RAD51D").
The introduction has to be expanded: Which are the genes other than BRCA associated with hereditary breast cancer? Are they high or moderate penetrant genes? And what about the data available in literature up to now regarding clinical and histopathological characteristics?
In the paragraph "2.1 Study population" is necessary to specify what emerges in the paragraph "3.1 Associations between clinico-pathological data and mutation status", in detail why patients presented for breast US and the absence of an organized screening. Moreover, when did patients undergo multigene panel testing?
Among patients who "presented for breast US by the means of an oppotunistic screening" (line 131), how many of them were aware to have a family history positive for breast cancer? This could explain why the carriers group had a signficantly higher number of unifocal tumor, which could represent a kind of "early diagnosis".
The higher histologic grade that is statistically associated with the carriers group could be due to the carriers of pathogenic variants in BRCA1, PALB2, RAD51C/D, which are approximatly one third of the entire group. Please, can you verify this possible association?
In Table 1, TNM stage is reported to significantly differ among carriers and non-carriers: however, this data is not cited in the text and it results of difficult interpretation because stages are excessively subdivided. It could be more informative to divide patients among early, locally advanced or metastatic stage.
Reviewer 3 Report
The manuscript is well written and scientifically sound and can be published after minor revisions:
Is “negative group” and “non-carriers” the same? The authors use both definitions throughout the manuscript and it’s confusing. Could it be specified “mutation-negative” group?
Why are tables with patients description marked “non published” in the file name? Are they not going to be published? They are part of the manuscript anyways, as I understand?
Lines 263-265: TP53 and ATM carriers only have 3 patients in each group. That’s not substantial amount to suggest they only develop ER- / ER+ tumors, respectively. Would require larger group so that some statistical analysis can be done, to assume that.
